# Differences in Cardiac-Pulsatility-Induced Displacement and Geometry Changes between the Cook ZBIS and Gore IBE: Postoperative Comparison Using ECG-Gated CTA Scans

**DOI:** 10.3390/diagnostics13030496

**Published:** 2023-01-29

**Authors:** Jaimy A. Simmering, Majorie van Helvert, Joost A. van Herwaarden, Cornelis H. Slump, Robert H. Geelkerken, Michel M. P. J. Reijnen

**Affiliations:** 1Department of Surgery, Division of Vascular Surgery, Medisch Spectrum Twente, 7512 KZ Enschede, The Netherlands; 2Multi-Modality Medical Imaging (M3i) Group, Faculty of Science and Technology, Technical Medical Centre, University of Twente, 7522 NB Enschede, The Netherlands; 3Department of Surgery, Division of Vascular Surgery, Rijnstate Hospital, 6815 AD Arnhem, The Netherlands; 4Department of Vascular Surgery, University Medical Center Utrecht, 3584 CX Utrecht, The Netherlands; 5Robotics and Mechatronics (RaM) Group, Faculty of Electrical Engineering, Mathematics and Computer Science, Technical Medical Centre, University of Twente, 7522 NB Enschede, The Netherlands

**Keywords:** IBD, ECG-gated CT, geometry, conformability, dynamic

## Abstract

To what extent the stentgraft design of iliac branch devices (IBDs) relates to dynamic deformation is currently unknown. Therefore, this study aimed to quantify and compare displacement and geometry changes during the cardiac cycle of two common IBDs. This paper presents a two-center trial with patients treated with a Zenith bifurcated iliac side (ZBIS) or Gore iliac branch endoprosthesis (IBE). All patients underwent a retrospective electrocardiogram (ECG)-gated computed tomographic angiography (CTA) during follow-up. Cardiac-pulsatility-induced displacement was quantified for the following locations: (neo) bifurcation of the aorta, IBD flow divider, distal markers of the internal iliac artery (IIA) component and first IIA bifurcation. Geometrical parameters (length, tortuosity index, curvature and torsion) were quantified over centerlines. Displacement was more pronounced for the IBE than the ZBIS, e.g., craniocaudal displacement of 0.91 mm (0.91–1.13 mm) vs. 0.57 mm (0.40–0.75 mm, *p* = 0.004), respectively. The IBDs demonstrated similar geometrical parameters in the neo-common iliac artery and distal IIA, except for the larger dynamic curvature and torsion of the distal IIA in IBEs. The IBEs showed more dynamic length and curvature change compared to the ZBIS in the stented IIA. The IIA trajectory showed more pronounced deformation during the cardiac cycle after placement of an IBE than a ZBIS, suggesting the IBE is more conformable than the ZBIS.

## 1. Introduction

Nowadays, an abdominal aortic aneurysm (AAA) is predominantly treated by endovascular aneurysm repair (EVAR) because of the preferable short- and mid-term outcomes compared to open repair [1]. In 20–40% of these patients, the AAA extends to one or both common iliac arteries (CIAs) [2,3,4]. In the case of such a concomitant CIA aneurysm, conventional EVAR is lacking an adequate distal landing zone in the CIA [4,5]. To assure effective distal sealing, the stentgraft can be extended into the external iliac artery (EIA) with subsequent intentional internal iliac artery (IIA) exclusion [6,7]. This procedure has been related to pelvic ischemia, which may lead to disabling buttock claudication (16–55%), erectile dysfunction (10–46%) or in rare cases spinal cord ischemia (0.3–0.8%) or colonic ischemia (0.5%) [6,8]. To prevent these complications, iliac branched devices (IBDs) were designed to exclude iliac aneurysms while preserving blood flow to the IIA. The first IBD reports describe the use of the Zenith bifurcated iliac side (ZBIS) branched device of the Zenith platform (Cook Medical, Bloomington, IN, USA), which is combined with balloon-expandable covered stents to seal into the IIA [9,10]. Experiences with the ZBIS showed the feasibility and safety of the IBD technique [11]. The iliac branch endoprosthesis (IBE) of the Gore Excluder platform (W.L. Gore & Associates, Flagstaff, AZ, USA) was designed for the same purpose as the ZBIS, but incorporates a complementary covered self-expanding IIA component. Both IBDs showed comparable short and midterm outcomes regarding technical success, freedom from type I or type III endoleaks and hypogastric patency [12,13,14]. Long-term results of the ZBIS include a five-year patency rate of 91.4% and a freedom of reintervention rate of 81.4% [11], while the five-year results of the IBE are still awaited.

In general, implantation of a stentgraft causes stiffening and straightening of the aortoiliac trajectories, especially in cases of high tortuosity [15,16,17]. Considering the tortuous nature of the iliac arteries, IBDs are thought to experience significant stresses and forces, while the stentgrafts are also continuously challenged by the blood pressure wave that is induced by the cardiac cycle. These forces introduce deformation, which may ultimately lead to complications and endanger long-term durability [18,19,20]. Stentgraft deformation may be defined as any change in the shape, geometry and/or size of the stentgraft over time and can be accurately quantified by dedicated electrocardiogram (ECG)-gated computed tomography (CT) analysis, which has proven to provide relevant clinical insight for aortoiliac stentgrafts [20,21,22,23,24]. Still, the deformation may differ between different stentgrafts developed for the same indication, due to differences in design. Stiffer stentgrafts are considered less conformable, which has been reported to increase the risk for clinical complications, due to a compliance mismatch between the target arteries and the stentgraft, while the risk for stent fatigue, in turn, is considered smaller in stiffer stentgrafts since the deformation is less in comparison to more flexible stentgrafts [25,26].

The stentgraft designs of the ZBIS and IBE differ, as well as the configuration of their IIA components [25]. The ZBIS consists of separate Z-shaped nitinol stent-rings with same-length struts sutured onto a Dacron graft material, while the IBE consists of partly overlapping spiral/Z-shaped nitinol stent-rings with different strut lengths encapsulated between an expanded polytetrafluoroethylene (ePTFE) graft material and a thin polymeric strip [27]. The Z-stents of the ZBIS are associated with lower flexibility and a tendency to kink in more angulated arteries [27]. Hence, the ZBIS and the balloon-expandable stents that are used as an IIA component for this IBD can be considered more stiff, compared to the relatively flexible IBE and accompanying self-expandable IIA components [25]. Potential differences in the dynamic deformation of stentgrafts may be related to the durability and applicability of these devices. The aim of this study was to quantitatively characterize and compare the cardiac-cycle-induced deformation of ZBIS and IBE stentgrafts.

## 2. Materials and Methods

### 2.1. Study Design and Population

This two-center trial enrolled patients with an isolated or concomitant iliac aneurysm who underwent elective treatment with an IBD between January 2011 and February 2019. The included patients were divided into two cohorts; a cohort of patients treated with a ZBIS and a cohort of patients treated with an IBE. All patients were treated according to the hospital’s standard practice and underwent a retrospective ECG-gated CTA scan at some point during follow-up. Approval was obtained from the ethical committee and the institutional review boards. The trial was registered on ClinicalTrials.gov (NCT03762525).

### 2.2. Image Acquisition

All scans were acquired with a 256-slice CT scanner (Brilliance iCT 256 scanner, Philips Healthcare, Eindhoven, The Netherlands). A total of 100 mL of contrast agent was intravenously administered at 5 mL/s. Scan acquisition was performed during the arterial phase, using bolus triggering at a threshold of 150 Hounsfield units in the distal descending thoracic aorta during a single inspiration breath hold after performing a standard breathing exercise. Scan parameters were as follows: tube voltage, 100–120 kV; current time product, 157–1551 mA∙s; slice thickness, 0.9–1.5 mm; slice increment, 0.4–1.0 mm; rotation time, 0.27–0.33 s; collimation, 128 × 0.625; pitch factor of 0.18 or 0.30, reconstruction matrix 512 × 512 pixels (iDose).

By means of retrospective gating, 8 or 10 equally sized phases of the cardiac cycle were obtained from 0% to 87.5% (12.5% intervals) or 0% to 90% (10% intervals) of the RR interval, respectively. Previous investigation has shown that displacement quantification on ECG-gated CT scans of 8- and 10-phase reconstructions can be accurately compared with an accuracy of 0.05 mm and interobserver variability of median 0.00 mm (−0.03 to 0.03 mm, intraclass correlation coefficient, ICC 0.839 *p* < 0.01) and an intraobserver variability of median 0.00 mm (−0.02 to 0.02 mm, ICC 0.853 *p* < 0.01) [15,24,28].

### 2.3. Image Processing

The image processing steps are described elsewhere [15,29,30]. In short, image registration was performed to obtain the phase-averaged mid-cardiac cycle CT volume, referred to as static CT volume from here on, and the deformation fields describing the displacement of each voxel for the individual phases with respect to the static CT volume.

For both IBDs, displacement amplitudes in the x-(lateral), y-(anterior–posterior) and z-(craniocaudal) directions were quantified by applying the deformation fields using backward mapping to points manually selected on the static CT volume. For each point, the pathlength during the cardiac cycle was calculated as well, i.e., the sum of the traveled distances of a point on the IBD between the subsequent phases of a cardiac cycle. The following seven points were selected: the bifurcation of the EVAR main graft (Points 1 and 2), the IBD flow divider (Points 3 and 4), the distal markers of the IIA component (5 and 6) and the first bifurcation of the IIA (7) (see Figure 1).

To quantify geometrical parameters, centerlines from the aorta to the IIA were obtained in Aquarius Intuition (version 18, TeraRecon, Inc., Foster City, CA, USA) on the static CT volume. By means of backward mapping using the deformation fields, the centerline of the static CT volume was converted to the centerlines of the individual phases. The centerlines were divided into the following 4 segments for further analysis (see Figure 2):Segment 1: main EVAR flow divider—IBD flow divider;Segment 2: IBD bifurcation—5 mm upstream of the end of the IIA component;Segment 3: 5 mm upstream of the end of the IIA component—5 mm downstream of the end of the IIA component;Segment 4: 5 mm downstream of the end of the IIA component—first native IIA bifurcation.

Several centerline parameters that are common to describe geometry of stentgrafts [15,17,25,29,31] were investigated for each centerline segment in both static and dynamic manner: length, tortuosity index (TI), curvature and torsion. Length was defined as the length of the centerline segment. TI was defined as the length of the centerline segment divided by the Euclidean distance between the start and end of this centerline segment. The dynamic length and TI change during the cardiac cycle were defined as the difference between the maximum and minimum values during the cardiac cycle. Curvature, a mathematical measure for the bending of a line [15,29], was calculated for each point on the centerlines. Torsion, a mathematical measure to quantify to what extent the curvature involves a third dimension, was also calculated for each point on the centerlines [15]. Since torsion can be either positive or negative, depending on the direction of the curve, the absolute torsion was used for further analysis. Static curvature and torsion were calculated for each point of the static centerline and the mean and maximum values were used for further comparison. Additionally, curvature and torsion were calculated for the centerlines of each cardiac phase, which was converted to a dynamic value for each centerline point by calculating the difference between the largest and smallest value for that point. This dynamic curvature and torsion were converted to a mean and maximum value for each centerline segment for further comparison.

### 2.4. Statistical Analysis

Continuous variables were presented as median (interquartile range [IQR]). Categorical variables were presented as number (percentage, %). Differences between the two groups were tested using a Mann-Whitney U test for continuous data and a Fisher exact or Pearson Chi-squared test (as appropriate based on the data) for categorical data. Differences in displacement between the selected points were tested with a one-way analysis of variance (ANOVA) with Bonferroni post-hoc testing, presented as mean difference (95% confidence interval, CI). Values of *p* < 0.05 were considered significant. Statistical analyses were performed using IBM SPSS statistics (version 28, IBM corporation, Armonk, NY, USA).

## 3. Results

In total, 32 patients were enrolled and underwent an ECG-gated CT scan at a median follow-up of 46 (42–83) days. The demographic characteristics are reported according to the SVS reporting standards [32] in Table 1. A total of 34 eligible IBDs, 17 ZBIS and 17 IBE, were implanted in the study group. The ZBIS IIA components were all balloon expandable Advanta V12^®^ covered balloon expandable stents (Getinge, Merrimack, NH, USA). The IBEs were predominantly accompanied by the complementary self-expandable IIA component, i.e., the hypogastric components (HGB). In four IBE cases, a self-expandable Viabahn (W.L. Gore & Associates, Flagstaff, AZ, USA) was used as an IIA component (extension) (see Figure 3).

Figure 4 shows more pronounced displacement patterns in the IBE group for all selected points, especially in the craniocaudal (z)-direction. This observation is supported by the statistically significant difference in maximum displacement amplitudes in all directions for all points, except in the lateral (x) displacement for the points located at the distal markers of the IIA component (Points 5 and 6) and in the anterior–posterior (y) displacement at the distal IIA component marker (point 6) and the first bifurcation of the IIA (Point 7) (Table 2). Thereupon, the pathlengths of all selected points were significantly larger in the IBE group, with a median difference of ~1 mm (Table 2). Furthermore, the displacement of the aorta/main device bifurcation was larger than for the distal IIA by 0.8 mm (95% CI 0.1–1.4 mm, *p* = 0.009) for the pathlengths and 0.2 mm (95% CI 0.1–0.3 mm, *p* < 0.001), 0.2 mm (95% CI 0.0–0.3 mm, *p* = 0.009) and 0.4 mm (95% CI 0.1–0.6 mm, *p* < 0.001) for the displacement in x-, y- and z-directions, respectively.

Considering the geometrical centerline parameters (Table 3), the CIA trajectories (Segment 1) demonstrated similar outcomes, both static and dynamic, among the two groups. Along the IIA component (Segment 2) and in the transition segment from the IIA component to the native IIA (Segment 3), the IBE showed more dynamic length and curvature change compared to the ZBIS. In addition, the static length and mean torsion of Segment 2 were larger in the IBE group. Similar to Segment 1, the trajectory of the native IIA downstream of the IIA components (Segment 4) demonstrated comparable static and dynamic geometrical parameters for the two IBD groups, except for the larger dynamic maximal curvature and mean and maximal torsion observed for the IBE group.

## 4. Discussion

The present study characterized and compared the dynamic deformation of two IBD stentgrafts by quantification of cardiac-pulsatility-induced displacement and changes in geometry based on multiphasic ECG-gated CT. The static geometries of the ZBIS and IBE were comparable, except for the IIA component segment (Segment 2) that is longer and experiences more torsion in the IBE group. The pulsatile displacement during the cardiac cycle was larger in the IBE than in the ZBIS throughout the aorto-iliac trajectory, especially in the craniocaudal (z) direction. The dynamic geometrical changes were also more pronounced in the IBE group for the iliac segments with a larger dynamic change in length (axial strain), curvature and torsion in the IBE group for the IIA segment (Segment 2) and the 10 mm around the end of the IIA component (Segment 3). The native IIA downstream of the stentgraft showed more dynamic curvature and torsion change in the IBE group. This may be explained by the IIA component in the IBE being longer and experiencing more axial strain, which progresses the pulsatile deformation stronger down the IIA in the IBE than in the ZBIS. The CIA segment (Segment 1) that runs from the aorta or EVAR bifurcation to the IBD bifurcation did not show dynamic geometry differences between the stentgrafts.

The findings in this study are consistent with the hypothesis that the stent design of the IBE is more flexible and thereby more conformable than the ZBIS [25]. The conformable and flexible IIA component of the IBE suggests a smaller compliance mismatch between native and stented IIA, which could benefit the long-term durability of the treatment [33,34,35]. In addition, the transition segment between the stent and the outflow vessel may be compromised by the transition of a stiff stent into a flexible artery. This, in turn, might lead to complications such as intimal damage reflected by stenosis and/or occlusion. Even though this could not be confirmed with the current results based on a small cohort with only one follow-up moment available, it would be of interest in future research.

To our knowledge, Schiava et al. [25] are thus far the only group to investigate the conformability and geometrical differences between the ZBIS and IBE. Their comparative analysis found no significant modification of the length and tortuosity index of the total iliac axes after IBE implantation, whereas significant modifications were found in patients treated with the ZBIS. This was suggested to be due to the less conformable design of the ZBIS compared to the IBE [25]. However, they did not compare the postoperative situations of IBE and ZBIS (only the pre- to postoperative change), and, more importantly, they only analyzed static CT scans. The influence of the cardiac cycle was not considered, while stentgrafts are continuously challenged by the consequent pulsatile displacement and forces. In a previous study, our group investigated the influence of IBE placement on the dynamic behavior and geometry of the CIA and IIA by analyzing pre- and postoperative ECG-gated CT scans in a multicenter observatory study [15]. This study concluded that placement of an IBE stentgraft stiffens and straightens the CIA-IIA trajectory. This warrants even more for the consequences of the smaller postoperative deformation and thus conformability of the ZBIS considering the long-term durability of the treatment. Still, the decreased deformation of the ZBIS configuration must be weighed against the potential metal fatigue of more conformable and flexible stent material, such as the IBE platform, which might lead to increased stent loading that could cause stent fracture in later stages.

The two groups demonstrated similar geometric behavior for the centerline through the (neo) CIA, i.e., from the EVAR/aorta bifurcation to the IBD bifurcation (Segment 1), while the pulsatile displacement in this location was larger in the IBE group. This might relate to the EVAR stentgrafts that accompanied the IBDs. The ZBIS was predominantly combined with the Zenith (Cook Medical, Bloomington, IN, USA), while the IBE was predominantly combined with the Gore Excluder (W.L. Gore & Associates, Flagstaff, AZ, USA). Similar to their IBDs, the EVAR stentgrafts of these manufacturers differ in conformability as well: the Zenith is considered relatively stiff, while the Excluder is considered relatively flexible [27]. Therefore, the Excluder may be more adaptive to the pulsatile motion induced by the blood pressure wave, which reflects as more displacement during the cardiac cycle [36]. The similar geometrical behavior at this level may relate to CIA geometry. Preoperatively, the CIA dimensions did not differ between the groups and these trajectories are relatively straight, which may explain why the geometry is not affected significantly by the blood pressure wave.

Due to the limited number of patients evaluated in this study, no claims were made regarding the clinical outcome. However, the results did show significant differences that may point towards potential clinical implications. What is also important to consider is that blood pressure may also play a role in the dynamic behavior of stentgrafts. The blood pressure was not obtained during the CT scans in this study. In future research, blood pressure measurements during the cardiac-gated CT scanning should be included since it can provide additional pivotal information regarding the magnitude of displacement and dynamic geometric changes [37]. Still, hypertension was observed more in the ZBIS group, while the pulsatile deformation was less in this group.

## 5. Conclusions

The IIA trajectory showed more pronounced displacement and geometry changes during the cardiac cycle after placement of an IBE than after placement of a ZBIS. This suggests that the IIA component of the IBE is more conformable than that of the ZBIS.

## Figures and Tables

**Figure 1 diagnostics-13-00496-f001:**
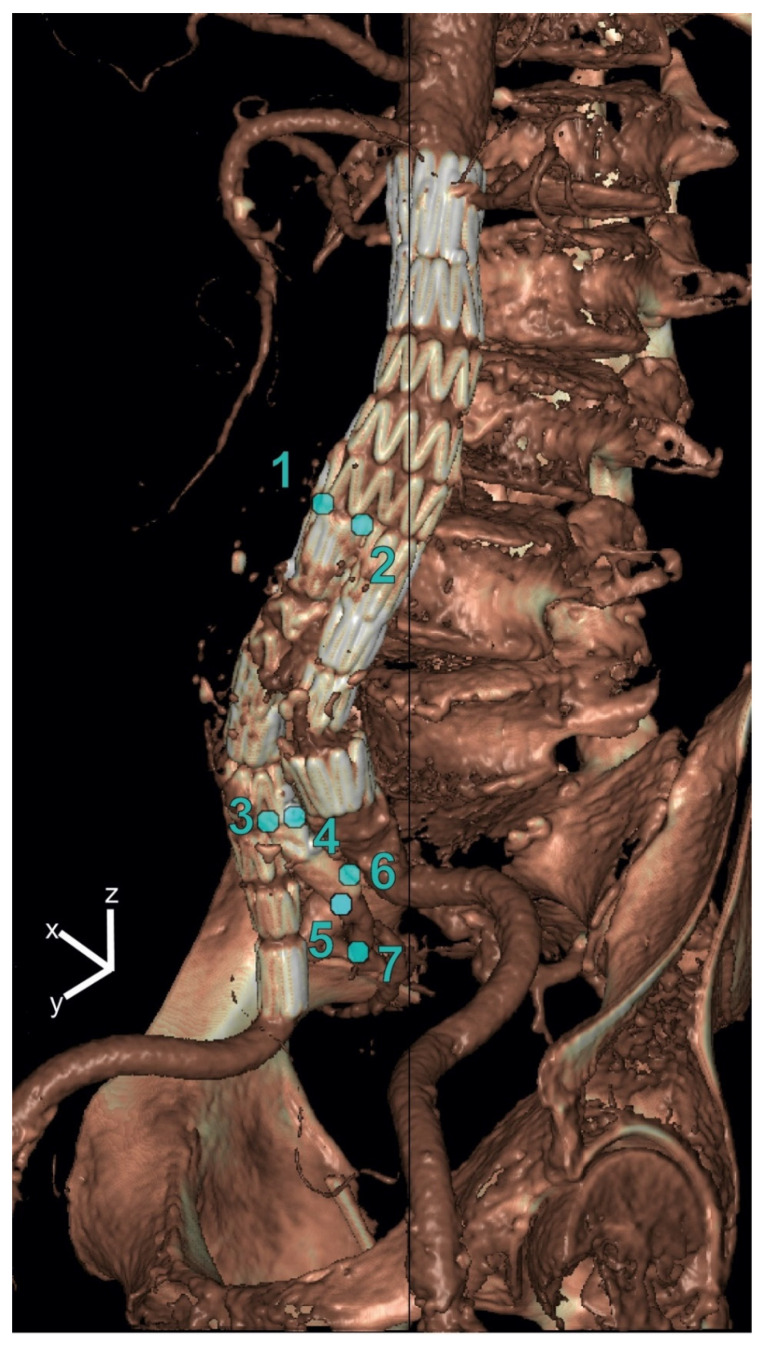
Selected points on the iliac branched device for displacement analysis in x-(lateral), y-(anterior–posterior) and z-(craniocaudal) directions. The points were selected at the bifurcation of the main graft (Points 1 and 2), the IBD flow divider (Points 3 and 4), the distal markers of the IIA component (Points 5 and 6) and the first bifurcation of the IIA (Point 7).

**Figure 2 diagnostics-13-00496-f002:**
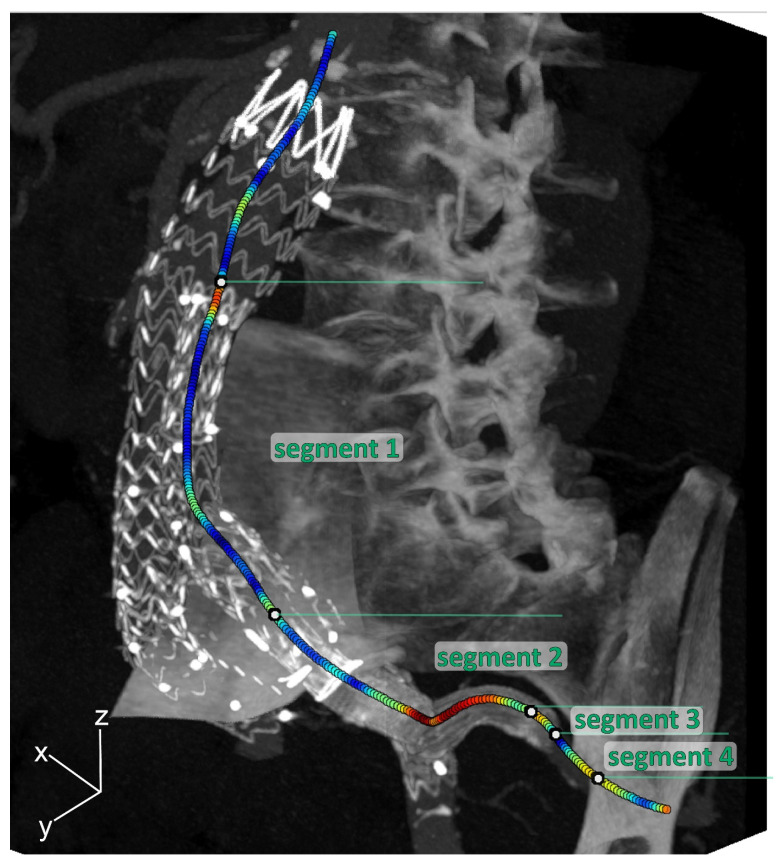
Three-dimensional maximal intensity projection of the mid-cardiac cycle volume example, including curvature color-coded centerlines (red is high curvature; blue is low curvature) for the internal iliac artery (IIA), which was divided into 4 segments: Segment 1: main EVAR flow divider—IBD flow divider; Segment 2: IBD bifurcation—5 mm upstream of the end of the IIA component; Segment 3: 5 mm upstream of the end of the IIA component—5 mm downstream of the end of the IIA component; Segment 4: 5 mm downstream of the end of the IIA component—first native IIA bifurcation.

**Figure 3 diagnostics-13-00496-f003:**
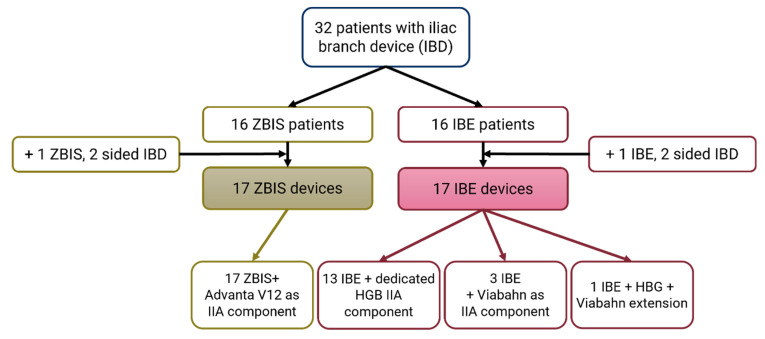
Flowchart of the included iliac branch device (IBD) patients. The Cook Medical Zenith bifurcated iliac side (ZBIS) was in all cases combined with the Advanta V12 balloon-expandable stent as internal iliac artery (IIA) component. The Gore Excluder platform iliac branched endoprosthesis (IBE) was combined with the accompanying self-expandable IIA component (HGB) in 13 of the 17 of the cases; the other cases had a self-expandable Viabahn as (additional) IIA component.

**Figure 4 diagnostics-13-00496-f004:**
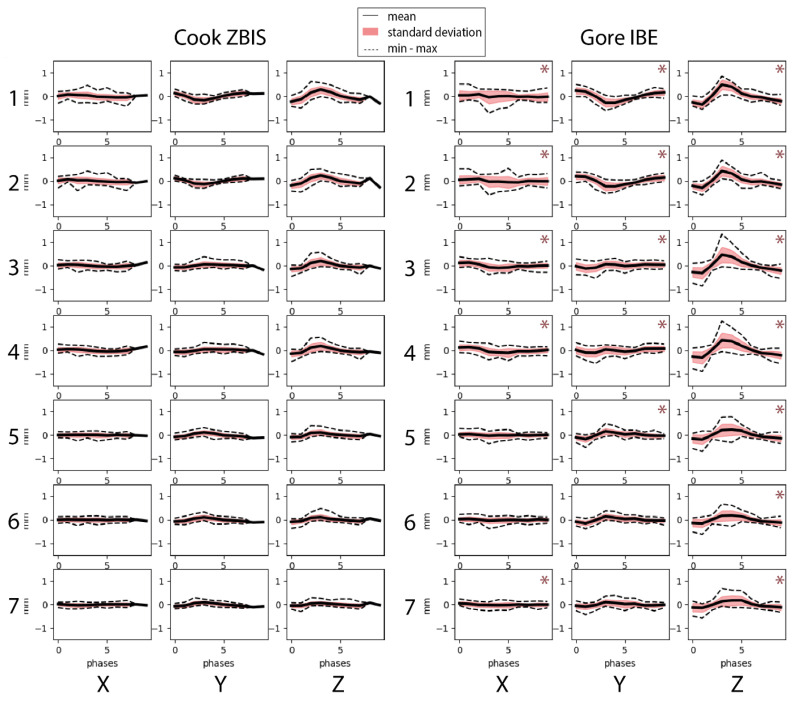
Displacement amplitudes in *x*-(lateral), *y*-(anterior–posterior) and *z*-(craniocaudal) direction for the selected points on the Cook Medical Zenith bifurcated iliac side (ZBIS, left) and the Gore Excluder platform iliac branched endoprosthesis (IBE, right). The points were selected at the bifurcation of the main graft (Points 1 and 2), the IBD flow divider (Points 3 and 4), the distal markers of the IIA component (5 and 6) and the first bifurcation of the IIA (Point 7). * indicate larger displacement for that point for the IBE group than the ZBIS group in the corresponding direction.

**Table 1 diagnostics-13-00496-t001:** Patient, anatomical and procedural characteristics.

Patient Characteristics	ZBIS (*n* = 16)	IBE (*n* = 16)	*p*-Value
Age, years	71 (67—78)	69 (66—75)	0.080
Male sex	15 (94%)	15 (94%)	1.000
Cardiovascular risk factors (SVS grading system) [32]			
Diabetes mellitus type II	1 (6%)	- (0%)	1.000
Smoking history	6 (38%)	8 (50%)	0.722
Hypertension	15 (94%)	8 (50%)	0.015
Creatinine level			0.572
Elevated up to 2.4 mg/dL	1 (6%)	1 (6%)	
2.5–5.9 mg/dL	1 (6%)	- (0%)	
>6.0 mg/dL	- (0%)	1 (6%)	
Hyperlipidemia	5 (31%)	11 (69%)	0.076
Cardiac status			0.091
Asymptomatic with cardiac history ^a^	3 (19%)	1 (6%)	
Symptomatic ^b^	6 (38%)	2 (13%)	
Pulmonary status			0.484
Asymptomatic ^c^	16 (100%)	14 (88%)	
Mildly symptomatic ^d^	- (0%)	2 (13%)	
**Procedural characteristics**			
Primary procedure EVAR + IBD IBD only	12 (75%) 4 (25%)	16 (100%) - (0%)	0.101
Side of IBE right left both	8 (50%) 7 (44%) 1 (6%)	9 (56%) 6 (38%) 1 (6%)	0.934
Time between surgery and scan, days	45 (42–58)	53 (42–262)	0.491
**Vessel geometries, mm**			
Diameter infrarenal aortic neck,	25 (22–28)	22 (21–26)	0.358
Maximum diameter infrarenal aorta	49 (34–55)	31 (26–48)	0.188
Treated Iliac arteries			
Maximum diameter treated CIA, mm	34 (31–36)	35 (33–38)	0.245
Length treated CIA, mm	70 (56–84)	58 (48–83)	0.382
Maximum diameter treated IIA, mm	9 (8–12)	8 (7–10)	0.363
Maximum diameter treated EIA, mm	11 (10–13)	10 (10–11)	0.204

SVS, Society of Vascular Surgery; CIA, common iliac artery; EIA, external iliac artery; IIA, internal iliac artery; EVAR, endovascular aneurysm repair; IBD, iliac branched device, ZBIS, Zenith bifurcated iliac side iliac branched device (Cook Medical, Bloomington, IN, USA); IBE, iliac branch endoprosthesis (W.L. Gore & Associates, Flagstaff, AZ, USA). Data presented as median (IQR), or as number (%). ^a^ Asymptomatic but with either remote myocardial infarction by history (6 months), occult myocardial infarction by electrocardiogram, or fixed defect on dipyridamole thallium or similar scan. ^b^ Any one of the following: stable angina, no angina but significant reversible perfusion defect on dipyridamole thallium scan, significant silent ischemia (1% of time) on Holter monitoring, ejection fraction 25% to 45%, controlled ectopy or asymptomatic arrhythmia, or history of congestive heart failure that is now well compensated. ^c^ Asymptomatic, normal chest radiograph, pulmonary function tests within 20% of predicted. ^d^ Asymptomatic or mild dyspnea on exertion, mild chronic parenchymal radiograph changes, pulmonary function tests 65% to 80% of predicted.

**Table 2 diagnostics-13-00496-t002:** Maximum displacement amplitudes, i.e., total displacement, in *x*-(lateral), *y*-(anterior–posterior) and *z*-(craniocaudal) direction and the pathlengths of the selected points 1–7 for both the Cook Medical Zenith bifurcated iliac side (ZBIS) and the Gore Excluder platform iliac branched endoprosthesis (IBE) groups. The following points were selected: the bifurcation of the main graft (Points 1 and 2), the IBD flow divider (Points 3 and 4), the distal markers of the IIA component (5 and 6) and the first bifurcation of the IIA. *

	Lateral (*x*) Displacement (mm)			Anterior–Posterior (*y*) Displacement (mm)			Craniocaudal (*z*) Displacement (mm)			Pathlength (mm)		
	ZBIS (*n* = 17)	IBE (*n* = 17)	*p*-Value	ZBIS (*n* = 17)	IBE (*n* = 17)	*p*-Value	ZBIS (*n* = 17)	IBE (*n* = 17)	*p*-Value	ZBIS (*n* = 17)	IBE (*n* = 17)	*p*-Value
**Point** **1**	0.26 (0.25–0.36)	0.47 (0.39–0.54)	0.008	0.42 (0.21–0.53)	0.64 (0.38–0.81)	0.011	0.57 (0.40–0.75)	0.91 (0.81–1.13)	0.004	1.8 (1.4–2.1)	2.9 (2.7–3.3)	<0.001
**Point 2**	0.29 (0.21–0.35)	0.48 (0.44–0.65)	0.002	0.35 (0.17–0.49)	0.58 (0.37–0.68)	0.012	0.48 (0.33–0.58)	0.78 (0.58–0.87)	0.004	1.6 (1.3–1.7)	2.8 (2.3–3.4)	<0.001
**Point 3**	0.29 (0.17–0.33)	0.38 (0.28–0.59)	0.022	0.21 (0.13–0.34)	0.43 (0.31–0.53)	0.009	0.40 (0.14–0.58)	0.67 (0.54–0.92)	0.002	1.4 (1.0–1.7)	2.4 (2.1–2.9)	<0.001
**Point 4**	0.27 (0.18–0.30)	0.46 (0.36–0.58)	<0.001	0.20 (0.13–0.31)	0.43 (0.31–0.56)	0.001	0.39 (0.16–0.56)	0.70 (0.54–0.82)	0.002	1.4 (0.9–1.8)	2.4 (2.3–2.8)	<0.001
**Point 5**	0.22 (0.16–0.29)	0.24 (0.20–0.39)	0.205	0.31 (0.17–0.35)	0.36 (0.31–0.50)	0.024	0.19 (0.16–0.38)	0.45 (0.32–0.70)	0.002	1.2 (0.9–1.3)	2.1 (1.8–2.5)	<0.001
**Point 6**	0.22 (0.15–0.25)	0.24 (0.23–0.35)	0.053	0.27 (0.14–0.35)	0.34 (0.25–0.51)	0.079	0.18 (0.15–0.28)	0.42 (0.28–0.55)	0.002	1.0 (0.8–1.2)	1.9 (1.7–2.2)	<0.001
**Point 7**	0.18 (0.12–0.24)	0.29 (0.20–0.32)	0.007	0.25 (0.14–0.30)	0.36 (0.20–0.43)	0.067	0.21 (0.13–0.27)	0.33 (0.27–0.57)	0.002	0.9 (0.8–1.2)	1.8 (1.6–2.7)	<0.001

* continuous variables are presented as median (interquartile range, IQR).

**Table 3 diagnostics-13-00496-t003:** Static and dynamic change of centerline parameters (length, tortuosity index [TI], mean curvature, maximum curvature, mean torsion and maximum torsion) for each centerline segment of the iliac branched devices (IBD): the Cook Medical Zenith bifurcated iliac side (ZBIS) and the Gore Excluder platform iliac branched endoprosthesis (IBE) groups: Segment 1, main EVAR flow divider to IBD flow divider; Segment 2, IBD flow divider to 5 mm upstream of the end of the internal iliac artery (IIA) component; Segment 3, 5 mm upstream to 5 mm downstream of the end of the IIA component; Segment 4, 5 mm downstream of the end of the IIA component to the first native IIA bifurcation. *

	Segment 1			Segment 2			Segment 3			Segment 4		
	ZBIS (*n* = 17)	IBE (*n* = 17)	*p*-Value	ZBIS (*n* = 17)	IBE (*n* = 17)	*p*-Value	ZBIS (*n* = 17)	IBE (*n* = 17)	*p*-Value	ZBIS (*n* = 17)	IBE (*n* = 19)	*p*-Value
Length—static (mm)	102.2 (95.2–114.2)	96.3 (80.2–132.4)	0.540	42.1 (27.6–46.0)	56.8 (55.1–64.3)	<0.001	9.9 (9.8–10.0)	9.9 (9.8–10.0)	0.658	17.9 (11.2–27.2)	17.5 (7.1–23.1)	0.838
Length—dynamic change (mm)	0.39 (0.22–0.62)	0.52 (0.39–0.60)	0.160	0.30 (0.22–0.35)	0.40 (0.33–0.56)	0.009	0.08 (0.07–0.09)	0.12 (0.10–0.16)	0.001	0.14 (0.07–0.20)	0.19 (0.13–0.25)	0.150
TI—static	1.072 (1.035–1.099)	1.079 (1.023–1.110)	0.634	1.064 (1.28–1.119)	1.068 (1.024–1.156)	0.518	1.021 (1.013–1.032)	1.018 (1.011–1.032)	0.658	1.037 (1.015–1.149)	1.025 (1.010–1.113)	0.734
TI—dynamic change	0.003 (0.002–0.004)	0.002 (0.001–0.003)	0.394	0.002 (0.001–0.003)	0.002 (0.002–0.004)	0.170	0.001 (0.000–0.001)	0.001 (0.001–0.002)	0.140	0.001 (0.000–0.006)	0.002 (0.000–0.004)	0.865
Mean curvature—static (m^−1^)	22.6 (19.4–26.4)	24.3 (21.0–28.3)	0.474	41.4 (33.1–50.3)	31.7 (24.0–37.4)	0.053	62.9 (50.5–86.5)	73.1 (53.4–85.1)	1.000	57.7 (46.6–86.9)	50.2 (33.4–70.6)	0.433
Mean curvature—dynamic change (m^−1^)	1.1 (1.0–1.5)	1.2 (1.1–1.3)	0.518	1.2 (0.9–1.5)	1.4 (1.2–2.0)	0.026	1.6 (0.9–2.1)	2.4 (2.0–3.1)	0.004	1.4 (1.2–2.1)	1.9 (1.5–3.1)	0.073
Maximal curvature—static (m^−1^)	78.7 (56.9–93.8)	63.5 (52.6–67.9)	0.140	79.8.1 (52.1–113.7)	72.9 (49.1–90.4)	0.496	89.1 (65.6–123.3)	96.2 (69.5–118.9)	0.683	93.7 (72.6–123.4)	90.5 (53.8–136.4)	0.708
Maximal curvature—dynamic change (m^−1^)	2.6 (2.1–3.9)	2.5 (2.1–3.2)	0.433	2.2 (1.7–3.0)	3.7 (2.6–4.6)	0.029	2.3 (1.2–2.6)	3.4 (2.6–4.0)	0.004	2.1 (1.8–2.5)	3.2 (2.4–7.1)	0.034
Mean torsion—static (m^−1^)	96.7 (83.1–110.2)	81.0 (65.8–103.6)	0.274	99.1 (77.6–117.7)	69.5 (64.9–76.9)	0.009	45.7 (32.5–60.3)	54.5 (23.7–107.2)	0.658	72.2 (59.7–94.3)	95.9 (80.8–130.0)	0.079
Mean torsion—dynamic change (m^−1^)	41.6 (14.3–99.5)	34.8 (15.4–78.7)	0.973	6.3 (3.7–11.6)	9.8 (5.9–12.4)	0.160	2.2 (1.4–3.1)	3.3 (2.5–4.6)	0.041	2.6 (2.0–3.4)	6.8 (4.2–22.5)	<0.001
Maximal torsion—static (m^−1^)	992.2 (645.9–1392.5)	918.1 (583.1–2337.3)	0.865	660.5 (428.9–769.7)	405.6 (314.6–547.2)	0.182	99.5 (53.0–117.2)	62.2 (49.8–210.4)	0.973	197.1 (135.1–364.6)	265.1 (164.7–590.9)	0.245
Maximal torsion—dynamic change (m^−1^)	1926.4 (143.7–4092.6)	1049.5 (189.6–4037.7)	0.919	81.1 (28.2–103.0)	80.8 (36.6–169.7)	0.496	3.7 (3.0–7.6)	8.3 (4.9–14.6)	0.106	6.0 (4.8–12.5)	19.9 (9.6–155.1)	0.014

* continuous variables are presented as median (interquartile range, IQR).

## Data Availability

The data used in this study may be requested via the corresponding author.

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
