# Peer review of "Differences in Cardiac-Pulsatility-Induced Displacement and Geometry Changes between the Cook ZBIS and Gore IBE: Postoperative Comparison Using ECG-Gated CTA Scans"

_diagnostics, 2023, doi:10.3390/diagnostics13030496_

Round 1

Reviewer 1 Report

The authors try to characterize the deformation of the iliac stent prosthesis based on dynamic imaging. The methodology of the study is based on high resolution of ECG triggered CT angiography and attempts to capture fine interaction of the circulatory system on the prostheses, so that the work provides innovative and potentially clinically relevant conclusions and deserves publication. However, the work deserves a number of improvements:

From Line 9: Term "motion" is used throughout the work? Is this "strain", "displacement"? Please describe?

Line 67 – 74: In the objective of the work, no direct link between the stiffness of the prosthesis and potential complications is established after detailed clinical introduction. Please describe precisely 

Line 71 – 72: The authors refer to the differences in the dynamic behavior of stent prostheses. However, there is no technical reference to the construction of the prosthesis and no technical properties of the stent prostheses.

Methodology: Has the displacement of the individual components compared to other components been recorded?

Table 2: The proven differences in displacement were estimated in direction x and y for about 0.5 mm, cranio-caudal in median < 1 mm. This raises the question of the reliability of the measurements. What measures have been taken to reduce variability? Were the repeated measurements performed and averaged to reduce intra- and interobserver variability?

Line 242 – 251 relate the rare complications after IBD to the properties of the prostheses, whereby no direct connection can be established, therefore the clinical data are unfairly mistreated for the justification of one's own goals. In lines 251 – 257  a connection between the properties of stent prostheses and functional/pathological consequences in the connecting arteries is established for the first time. For lines 242 – 251, I recommend the excurse about the interaction between of stent/stent graft and vessel geometry.

Line 256 - 257: When comparing the clinical differences between ZBIS and IBE in terms of prosthesis compliance, the authors propose a future analysis. With a small number of IBDs performed worldwide, low complication rate and "motion" < 1 mm, what would be the calculated number of cases for the study?

I would like to thank the authors for a great study, I look forward to improved manuscript and wish further scientific inspiration.

Author Response

Please see the attachtment.

Reviewer 2 Report

Thank you for reporting about ‘Differences in cardiac-pulsatility-induced motion and geometry changes between the COOK ZBIS and Gore IBE: Postoperative comparison using ECG-gated CTA scans. ‘

This paper contains valuable information for IIA reconstruction devices. I have some concerns about your report.

1. ECG gated CT imaging is being taken; how much is the effect of IIA on cardiac cycle?

2. The Fig 4 was not found. (Page 5, line 170)
